# Entropic Dynamics of Stocks and European Options

**DOI:** 10.3390/e21080765

**Published:** 2019-08-06

**Authors:** Mohammad Abedi, Daniel Bartolomeo

**Affiliations:** Department of Physics, University at Albany-SUNY, Albany, NY 12222, USA

**Keywords:** maximum entropy method, entropic dynamics, geometric Brownian motion, European options, Black–Scholes model, Black–Scholes–Merton equation, put-call parity

## Abstract

We develop an entropic framework to model the dynamics of stocks and European Options. Entropic inference is an inductive inference framework equipped with proper tools to handle situations where incomplete information is available. The objective of the paper is to lay down an alternative framework for modeling dynamics. An important information about the dynamics of a stock’s price is scale invariance. By imposing the scale invariant symmetry, we arrive at choosing the logarithm of the stock’s price as the proper variable to model. The dynamics of stock log price is derived using two pieces of information, the continuity of motion and the directionality constraint. The resulting model is the same as the Geometric Brownian Motion, GBM, of the stock price which is manifestly scale invariant. Furthermore, we come up with the dynamics of probability density function, which is a Fokker–Planck equation. Next, we extend the model to value the European Options on a stock. Derivative securities ought to be prices such that there is no arbitrage. To ensure the no-arbitrage pricing, we derive the risk-neutral measure by incorporating the risk-neutral information. Consequently, the Black–Scholes model and the Black–Scholes-Merton differential equation are derived.

## 1. Introduction

In pursuit of understanding and describing phenomena, scholars often encounter situations in which information about the subject of interest is limited. Entropic inference is an inductive inference framework equipped with proper tools to handle situations where incomplete information is available [1,2,3]. Such tools are probability theory, relative entropy, and information geometry. To give a quantitative description of the state of partial knowledge, we use the probability distribution. Once new information is available, we can update our state of partial knowledge by maximizing the relative entropy. It is of crucial importance to notice that this notion of entropy does not originate from physics; however, it is the other way around. The celebrated notion of entropy in physics is indeed an example of the notion of relative entropy in entropic inference [3].

The information about the subject matter, the dynamical variable, takes the form of constraint. We use the method of Maximum Entropy, specifically the relative entropy, to incorporate the information available to answer questions about the system of interest. The Maximum Entropy method is designed to reflect the appealing property of Principle of Minimal Updating. The Principle of Minimal Updating ensures that the probability distribution is updated to the extent required by the new information.

Due to incomplete information, addressing a question renders a probabilistic answer, namely a probability distribution. The relative entropy is designed to update the state of partial knowledge, namely the probability distribution, whenever a new piece of information is available. The advantage of an entropic framework is the flexibility with which it can be adapted to deal with a variety of situations: once one realizes how information is codified into constraints, it is straightforward to modify the constraints to create new models. The main challenge is to identify the correct variables and the information that happens to be relevant to the problem.

Entropic Dynamics, ED, is an example of entropic inference framework in which the dynamical theories are modeled. Inference framework does not provide a notion of time, namely an instant and a duration or clock. These notions need to be appended to enable inference framework to model dynamics where time plays an important role. In ED, an entropic notion of time [4] is introduced which is suited to the system of interest. We are exploiting *Mechanics without a Mechanism* so as to develop a purely inferential dynamics completely divorced from physics [5], for example, there is no energy or momentum conservation. Entropic Dynamics has been extensively used to model the dynamics of subatomic particles [5,6,7,8,9,10,11].

The practice of modeling the dynamics of stock goes back to Bachelier’s thesis [12] which predates Einstein work on Brownian Motion [13]. Bachelier modeled the dynamics of stock price using the stochastic process and he came up with what is known as Geometric Brownian Motion. Bachelier’s work was rediscovered by Samuelson [14]. The models were developed with the assumption of no jump [14,15]. The Geometric Brownian Motion model was used by Black and Scholes to value Options [16,17]. The dynamics of stocks and pricing of Options were further developed by Merton to include jumps [18]. Numerous extensions and applications were proposed such as introducing stochastic volatility [19,20,21,22,23,24,25,26,27,28]. Our model differs from the stochastic process approach since we do not make an assumption about the process, but we derive the dynamics of price i.e., GBM. Entropic modeling complements the stochastic process modeling. An advantage of Entropic Dynamics is that it can unify the models developed in different branches of science.

In this work, we model the dynamics of stock price. The stock price is changing and there are other factors affecting the price to change. Due to a lack of information, we cannot keep track of the exact factors affecting the price; instead, based on the information available to us, we can proceed to predict the distribution of the price providing change happens. We do not take into account that jumps can happen; we develop our model in the regime of continuous evolution. The ED of the stock model yields the Geometric Brownian Motion dynamics. This is the new contribution where we *derive* the assumption of stock price undergoing a Geometric Brownian Motion. Furthermore, we address the question of how the probability density distribution changes over time which will be a Fokker–Planck equation.

The main objective this work is to lay down an alternative framework to model dynamics. While the main focus of other articles on the subject are to assume different stochastic models for the dynamics of stock price, we derive the stochastic dynamic model from our formalism. This is where we contribute to the literature on the subject by introduction a formalism to derive dynamics. A simple comparison can be made with the Newton’s theory of motion where the dynamics are assumed, the Newton’s laws, and the Lagrangian formalism where the dynamics can be derived.

Next, we proceed to apply the entropic stock model to price European Options. To do so, we construct the risk-neutral probability distribution, applying which to value Options ensures the no-arbitrage pricing. Deriving the risk-neutral distributions is done within the Entropic Dynamics formalism, i.e., we use the risk-neutral information to derive the appropriate measure. We will arrive at the Black–Scholes, BS, model [16,17] by assuming that the volatility and risk free rate are constant over time. The celebrated Black–Scholes-Merton, BSM, differential equation [18] is derived by taking the time derivative of the expected payoff at maturity.

## 2. Entropic Stock Model

We want to model the dynamics of stock’s price. Given the current price of the stock, how will the price change? In this work, we do not need to address how we price or value the stock, namely how market determines the price, but instead, assuming that is valued, we would like to come up with a model to describe how the price would change.

### 2.1. Scale Invariant Measure

Specifying the subject matter, the underlying dynamical variable to model, is of crucial importance. Prior information about the dynamical variable provides guidance to specify the proper dynamical variable and the prior probability density.

The dynamics should manifest scale invariance symmetry. As we will see, the stochastic process we derive will be manifestly scale invariant. However, to have our framework be manifestly scale invariant, we try to construct the probability densities that are invariant under scale transformation. This choice of having a formalism to be manifestly invariant will lead to choosing a logarithm of price as the proper dynamical variable to model.

Where does this symmetry come from? How do we know the symmetry before we derive the model? The symmetry is due to the interest of investors in investing in the stocks with higher returns. For any investor, it is not important what the price is, but, if they invest in a stock, how much return they will receive from that investment. A stock with a low price could have a higher return compared with a stock with a higher price. This would make the stock with a higher return more favorable. The demand to purchase and invest in the stock with a higher return will increase, which, in turn, will lead to a rise in the price of the stock. This rise of the stock price will decrease the stock return. Supply and demand forces not only determine the market value of the stocks but most importantly they will equilibrate the return of the stocks such that all stocks, which have the same volatility, have the same return.

The scale invariant objects are the probabilities and the relative entropy and constraints. The probability densities need not be invariant under scaling transformation; however, to have a formalism that is manifestly invariant under scale transformation, we require the probability densities to be invariant under scale transformation. The scale transformation of the price is the following:(1)$$\tilde{S}=l\;S,$$where *l* is a positive constant scaling factor. Choosing the price of the stock as the dynamical variable would lead to a probability density distribution that is not invariant under scaling transformation. To have a manifestly invariant formalism, we would like to find a representation such that the probability densities are invariant under the scaling transformation. Mathematically, we are looking for a function of the price f(S) such that the probability densities are scalars:(2)$$P\left(f(\tilde{S})\right)=P\left(f(S)\right).$$Since probabilities are required to be scalar, then we have the following relation:(3)$$df\left(\tilde{S}\right)=df\left(S\right).$$This equation would lead to the following relation for f(S):(4)$$f\left(\tilde{S}\right)=f\left(S\right)+C,$$where *C* is independent of price. Using Equation (Equation 1), we get
(5)f(lS)=f(S)+C(l).
This relation would constrain the constant function C(l). Using the same equation for a different scaling factor,
(6)$$f\left(l^{\prime }lS\right)=f\left(lS\right)+C\left(l^{\prime }\right)=f\left(S\right)+C\left(l\right)+C\left(l^{\prime }\right)=f\left(S\right)+C\left(l^{\prime }l\right),$$
which results in
(7)$$C\left(l\right)+C\left(l^{\prime }\right)=C\left(l^{\prime }l\right).$$
The solution to Equation (Equation 7) is the following:(8)C(l)=lnl.Plugging Equation (Equation 8) in Equation (Equation 5) yields
(9)f(lS)=f(S)+lnl.
The unique solution to the above functional equation is the logarithm function,
(10)f(S)=lnS.
We can simply check that the scaling transformation of the price would shift the f(S) by a constant that satisfies the desired property of Equation (Equation 3). All in all, we conclude that choosing the microstate as the logarithm of the price would be beneficial i.e., the probability density distribution would be a scalar function under the scaling transformation of the price.

### 2.2. Statistical Model

Our subject matter is the logarithm of the price. Therefore, the question we wish to address is given the current logarithm of the price, lnS, how will the log price change? Given incomplete information about the subject, we can give a probabilistic answer $P(\ln S^{\prime }|\ln S)$ that is through the method of Maximum Entropy, ME. Assigning the distribution $P(\ln S^{\prime }|\ln S)$ requires utilizing the relative entropy,
(11)$$S\left[P,Q\right]=-\int d\ln S^{\prime }\;P\left(\ln S^{\prime }|\ln S\right)\;\ln\frac{P(\ln S^{\prime }|\ln S)}{Q(\ln S^{\prime }|\ln S)},$$
where $Q(\ln S^{\prime }|\ln S)$ is the prior probability distribution, and specifying it is the subject of the next section. The relative entropy is designed as a tool of inference to update the state of partial belief whenever new information is accessible [29,30]. It is noteworthy that the notion of entropy does not belong to physics. In the development of theoretical physics, it became manifest that entropy as a tool of inference can be used to model statistical physics where the thermodynamic entropy is shown to be derived from inference [31,32,33]. Maximizing the entropy, updating the distribution, with respect to no information but normalization constraint would yield a posterior distribution that is exactly the same as the prior. This is exactly what we expect; given no new information, we would have no reason to update our belief, the distribution.

#### 2.2.1. The Prior

The prior distribution presents the information at hand before any new information is accessed. The prior distribution can be found from maximizing the relative entropy,
(12)$$S\left[Q,\mu \right]=-\int d\ln S^{\prime }\;Q\left(\ln S^{\prime }|\ln S\right)\;\ln\frac{Q(\ln S^{\prime }|\ln S)}{\Omega (\ln S^{\prime }|\ln S)},$$
where $\Omega (\ln S^{\prime }|\ln S)$ reflects our belief when no information is available. In this case of extreme uncertainty where the stock’s price can undergo any change irrespective of the current price, we have to assign uniform distribution. However, a priori, we know that the log price would make a small change in vicinity of the current log price, namely the motion of log price is continuous. Such information is harnessed as the following constraint:(13)$${\langle {\left(\Delta \ln S\right)}^{2}\rangle }_{Q}={\langle {\left(\ln\frac{S^{\prime }}{S}\right)}^{2}\rangle }_{Q}=k.$$Maximizing the relative entropy Equation (Equation 12) subject to the prior information and normalization yields
(14)$$Q\left(\ln S^{\prime }|\ln S\right)=\frac{1}{\eta }\exp\left[-\frac{\alpha }{2}\;{\left(\ln\frac{S^{\prime }}{S}\right)}^{2}\right],$$
where α is a Lagrange multiplier that is large and will be specified later on and $\eta =\int d\ln S^{\prime }\;\exp\left[-\frac{\alpha }{2}\;{\left(\ln\frac{S^{\prime }}{S}\right)}^{2}\right]$. Notice that, for large α, the prior Equation (Equation 14) is a very sharp Gaussian distribution anchored at the current log price lnS. Next, we consider the interpretation of the Lagrange multiplier α.

#### 2.2.2. Volatility and Entropic Clock

Inference is not equipped with the notion of time; we need to add this notion to enable the entropic inference framework to model dynamics. The same as entropy, time is also considered as a quantity belonging to physics. Time in physics, the same as entropy, is an example of entropic time [4]. Here, we introduce an entropic notion of clock, a duration that is tailored to stock price dynamics. We introduce the notion of entropic clock as follows:(15)$$\alpha =\frac{1}{\sigma^{2}\;\Delta t},$$where $\sigma^{2}$ is the volatility of the stock log price. If the volatile happened to be constant, then this notion of clock resembles Newtonian time [6]. Entropic clock could be also defined in such a way to be the same as relativistic time [8,9,10]. Upon the system of interest, a relevant entropic time could be introduced, which would simplify the dynamics.

#### 2.2.3. The Constraints

In entropic inference framework, the information relevant to dynamical variable is introduced in the form of constraint. Notice that, if we take into account wrong information, we will get an incorrect model. The only piece of information pertaining to change of price is:(16)$${\langle \ln\frac{S^{\prime }}{S}\rangle }_{P}=k^{\prime },$$where $k^{\prime }$ will be determined shortly and *P* stands for the posterior transition density. Next, by Taylor expanding the log function to the second order, we get
(17)$$\ln\frac{S^{\prime }}{S}\approx \;\frac{\Delta S}{S}-\frac{1}{2}{\left(\frac{\Delta S}{S}\right)}^{2}$$
and then take the expectation with respect to the transition distribution,
(18)$${\langle \ln\frac{S^{\prime }}{S}\rangle }_{P}\approx \;{\langle \frac{\Delta S}{S}\rangle }_{P}-\frac{1}{2}{\langle {\left(\frac{\Delta S}{S}\right)}^{2}\rangle }_{P},\;$$
where the first term of the expansion is specified by a drift,
(19)$${\langle \frac{\Delta S}{S}\rangle }_{P}=\mu \;\Delta t.$$
In our model, we do not need to know what the value of the drift is; however, to apply the model, the drift is a piece of information that ought to be found. Each stock has its own drift that reflects the performance of the company. Another separate entropic model can be developed to take into account the fundamental ratios and relevant information of the company to derive the drift term.

To specify the second term of the expansion in Equation (Equation 18), we maximize the entropy subject to normalization and the constraint Equation (Equation 16). This will yield the transition probability:(20)$$P\left(\ln S^{\prime }|\ln S\right)=\frac{1}{\xi }\;\exp\left[-\frac{\alpha }{2}{\left(\ln\frac{S^{\prime }}{S}\right)}^{2}+\beta \;\ln\frac{S^{\prime }}{S}\right],$$where β is a Lagrange multiplier corresponding to the constraint Equation (Equation 16) and the normalization factor is $\xi =\int d\ln S^{\prime }\;\exp\left[-\frac{\alpha }{2}{\left(\ln\frac{S^{\prime }}{S}\right)}^{2}+\beta \;\ln\frac{S^{\prime }}{S}\right]$. We can rewrite the distribution Equation (Equation 20) as a Gaussian distribution in $\log S^{\prime }$,
(21)$$P\left(\ln S^{\prime }|\ln S\right)=\frac{1}{Z(\alpha ,\beta ,\ln S)}\;\exp\left[-\frac{\alpha }{2}{\left(\ln\frac{S^{\prime }}{S}-\frac{\beta }{\alpha }\right)}^{2}\right],$$
where the new normalization factor is $Z=\int d\ln S^{\prime }\;\exp\left[-\frac{\alpha }{2}{\left(\ln\frac{S^{\prime }}{S}-\frac{\beta }{\alpha }\right)}^{2}\right]$. We attain the Weiner process for the logarithm of price,
(22)$$\ln\frac{S^{\prime }}{S}=\langle \ln\frac{S^{\prime }}{S}\rangle +\Delta W,$$
(23)$$\langle \ln\frac{S^{\prime }}{S}\rangle =\beta \;\sigma^{2}\;\Delta t\;\;\langle \Delta W\rangle =0,\;\langle {\left(\Delta W\right)}^{2}\rangle =\frac{1}{\alpha }=\sigma^{2}\;\Delta t.$$
Now, to find the second term in Equation (Equation 18), ${\langle {\left(\frac{\Delta S}{S}\right)}^{2}\rangle }_{P}$, we square the Taylor expansion Equation (Equation 17) and then take the expectation with respect to transition distribution,
(24)$${\langle {\left(\ln\frac{S^{\prime }}{S}\right)}^{2}\rangle }_{P}={\langle {\left(\frac{\Delta S}{S}\right)}^{2}\rangle }_{P}=\sigma^{2}\;\Delta t.\;$$
All in all, we specify the constraint Equation (Equation 16) and the Lagrange multiplier β,
(25)$$\langle \ln\frac{S^{\prime }}{S}\rangle \approx \;\langle \frac{\Delta S}{S}\rangle -\frac{1}{2}\langle {\left(\frac{\Delta S}{S}\right)}^{2}\rangle =\mu \;\Delta t-\frac{1}{2}\;\sigma^{2}\;\Delta t=k^{\prime }$$
and $\frac{\beta }{\alpha }=\beta \;\sigma^{2}\;\Delta t=k^{\prime }=\mu \;\Delta t-\frac{1}{2}\;\sigma^{2}\;\Delta t$, thus we have $\beta =\frac{\mu }{\sigma^{2}}-\frac{1}{2}$. To summarize, we can rewrite the transition probability,
(26)$$P\left(\ln S^{\prime }|\ln S\right)=\frac{1}{Z}\;\exp\left[-\frac{1}{2\sigma^{2}\;\Delta t}{\left(\ln S^{\prime }-\left(\ln S+\mu \Delta t-\frac{1}{2}\sigma^{2}\Delta t\right)\right)}^{2}\right].$$
This is the normal distribution for the log price that leads to a Wiener process for the log price,
(27)$$\ln\frac{S^{\prime }}{S}={\langle \ln\frac{S^{\prime }}{S}\rangle }_{P}+\Delta W,$$
where
(28)$${\langle \ln\frac{S^{\prime }}{S}\rangle }_{P}=\mu \;\Delta t-\frac{1}{2}\;\sigma^{2}\;\Delta t\;\;{\langle \Delta W\rangle }_{P}=0$$
with the following fluctuation,
(29)$${\langle {\left(\Delta W\right)}^{2}\rangle }_{P}=\sigma^{2}\;\Delta t.$$
It is noteworthy that we can see why, in Taylor expansion, we kept the second order in Equations (Equation 17), (Equation 24). The higher order terms are proportional to higher order of Δt. Thus, in the regime of continuous motion, we only kept terms that converge to the left-hand side in probability. To rewrite the transition probability in terms of the price of the stock, we just transform from log price back to price using the following transformation:(30)$$d\ln S^{\prime }\;P\left(\ln S^{\prime }|\ln S\right)=dS^{\prime }\;\frac{1}{S^{\prime }}\;P\left(\ln S^{\prime }|\ln S\right)=dS^{\prime }\;P\left(S^{\prime }|S\right).$$Then, we get the probability of price,
(31)$$P\left(S^{\prime }|S\right)=\frac{1}{Z\;S^{\prime }}\exp-\frac{1}{2\sigma^{2}\;\Delta t}{\left[\ln S^{\prime }-\left(\ln S+\mu \;\Delta t-\frac{1}{2}\sigma^{2}\;\Delta t\right)\right]}^{2}.$$
This is the lognormal distribution for the $S^{\prime }$. Notice that this is the transition distribution for a very short time interval. Assuming that the drift μ and volatility are uniform; then, for a finite time interval *T*, we will get a lognormal distribution with Δt replaced by the finite time *T*. However, if we relax the assumption of uniformity, we will no longer end up with a lognormal distribution. The resultant distribution will be the solution of a Fokker–Planck equation that is the subject of the next section.

### 2.3. Entropic Instant and Probability Dynamics

To complete the notion of entropic time, we need to define entropic instant. An *entropic instant* is defined as
(32)$$p{\left(\ln S^{\prime }\right)}_{t^{\prime }}=\int d\ln S\;P\left(\ln S^{\prime }|\ln S\right)\;p{\left(\ln S\right)}_{t}.$$
The distribution $p{\left(\ln S\right)}_{t}$ represents all available information at an instant *t* and the next instant is defined as $p{\left(\ln S^{\prime }\right)}_{t^{\prime }}$. For simplicity, we write p(lnS,t) instead of $p{\left(\ln S\right)}_{t}$. This parameter *t* has a nice property of being ordered and having an arrow [4]. The integral Equation (Equation 32) can be written in a differential form:(33)$$\partial_{t}p\left(\ln S,t\right)=-\frac{\partial }{\partial \ln S}\left(\left(\mu -\frac{1}{2}\sigma^{2}\right)\;p\left(\ln S,t\right)\right)+\frac{1}{2}\;\frac{\partial^{2}}{\partial {\left(\ln S\right)}^{2}}\;\left(\sigma^{2}\;p\left(\ln S,t\right)\right).$$This is a Fokker–Planck equation that governs the dynamics of density distribution.

## 3. European Option Pricing

In this section, we use the entropic stock model to derive the risk-neutral probability density. Using the risk-neutral measure for valuation amounts to a no-arbitrage pricing. We simply value the options at maturity by its expected payoff using the risk-neutral measure and then the premium is calculated as the discounted expected payoff. Next, we derive the Black–Scholes-Merton partial differential equation governing the time evolution of the options premium following the same argument.

### 3.1. Black–Scholes Model: Risk Neutral Valuation

Derivative securities should be priced such that there is no arbitrage opportunity. To have a no-arbitrage pricing, we derive the risk-neutral probability density. Risk-neutral measure is derived by imposing the risk-neutrality constraint [34]:(34)$$\;\mu =r_{f},$$where $r_{f}$ is the risk free rate. To derive the risk-neutral measure, we follow the same procedure as we did to derive the distribution for the underlying security, but, instead, on the drift in Equation (Equation 19), we impose the constraint Equation (Equation 34). The resultant risk neutral probability density is as follows:(35)$$P\left(\ln S^{\prime }|\ln S\right)=\frac{1}{Z}\;\exp\left[-\frac{1}{2\sigma^{2}\;\Delta t}{\left(\ln S^{\prime }-\left(\ln S+r_{f}\Delta t-\frac{1}{2}\sigma^{2}\Delta t\right)\right)}^{2}\right].$$

To value the European Call option, we start with computing the expected payoff at maturity using the risk neutral probability assuming that the the risk free rate and the volatility are uniform, constant in time and independent of price. The expected payoff, denoted by $V_{c}$, at maturity is given by the difference between the expected sale price and the expected purchase price,
(36)$$V_{c}={\langle \mathrm{Sale}\rangle }_{LN,T}-{\langle \mathrm{Purchase}\rangle }_{LN,T},$$
where we have
(37)$$\begin{array}{cc} {\langle \mathrm{Sale}\rangle }_{T}= & \int_{K}^{\infty }\;dS\;P\left(S,T|S_{0}\right)\;S,\\ {\langle \mathrm{Purchase}\rangle }_{T}= & \int_{K}^{\infty }\;dS\;P\left(S,T|S_{0}\right)\;K, \end{array}$$
where *K* is the strike price and $S_{0}$ is the current stock price. We integrate from the strike price since, if the price is less than the strike price, we will not exercise the call option. Then, the expected payoff can be written as
(38)$$V_{c}=\int_{K}^{\infty }\;dS\;P\left(S,T|S_{0}\right)\;\left(S-K\right).$$
The Premium for call option is just the discounted value of the payoff. The second piece of information about the risk-neutral valuation is to discount the future values with the risk free rate [34]:(39)$$C=e^{-r_{f}T}\;V_{c}.$$Since we know the current price $S_{0}$ and we have already assumed the risk free rate and the volatility to be uniform, then the probability distribution at maturity is given by
(40)$$\begin{array}{cc} P(S_{T}|S_{0})= & \int \;d\tilde{S}\;P\left(S_{T}|\tilde{S}\right)\;P\left(\tilde{S}|S_{0}\right)=\int \;d\tilde{S},\;P\left(S_{T}|\tilde{S}\right)\;\delta \left(\tilde{S}-S_{0}\right),\\ ∼ & LN(\ln S_{0}+r_{f}\;T-\frac{\sigma^{2}\;T}{2}\;,\;\sigma \;\sqrt{T}). \end{array}$$
This is still a lognormal distribution of price for a finite time interval *T*. Notice that the nice lognormal distribution is achieved since the interest rate and volatility are time and price independent. Relaxing these assumptions would lead to a different distribution that is a solution of the Fokker–Planck equation. Calculating the expected sale price yields
(41)$${\langle \mathrm{Sale}\rangle }_{LN,T}=S_{0}\;\exp\left[rT\right]\;N\left(d_{1}\right),$$
where $d_{1}=\frac{\ln S_{0}+r_{f}T+\frac{\sigma^{2}T}{2}-\ln K}{\sigma \;\sqrt{T}}$ and $N(d_{1})$ is the standard normal cumulative distribution function
(42)$$N\left(d_{1}\right)=\frac{1}{\sqrt{2\pi }}\int_{-\infty }^{d_{1}}dx\;e^{-\frac{x^{2}}{2}}.$$
In addition, the expected purchase price can be computed,
(43)$${\langle \mathrm{Purchase}\rangle }_{LN,T}=KN\left(d_{2}\right),$$
where $d_{2}+\sigma \;\sqrt{T}=d_{1}$. Then, the premium of call option is
(44)$$C=S_{0}\;N\left(d_{1}\right)-e^{-r_{f}T}\;K\;N\left(d_{2}\right).$$
This is the Black–Scholes model for a European call option. We can follow the same logic to value a European put option. The expected payoff at the maturity for a put option is
(45)$$V_{p}=\int_{0}^{K}\;dS\;P\left(S,T|S_{0}\right)\;\left(S-K\right).$$
Notice that we integrate from zero to the strike price *K* because, if the price is greater than the strike price, we will not exercise the put option. Discounting this expected payoff will yield the put premium,
(46)$$P=e^{-r_{f}T}\;K\;N\left(-d_{2}\right)-S_{0}\;N\left(-d_{1}\right).$$
We can simply check that the call and put premium satisfy the so-called call-put parity relation,
(47)$$C-P=e^{-r_{f}T}\;\left(F-K\right),$$
where *F* is the forward price, $S_{0}=e^{-r_{f}T}F$. Therefore, we can conclude that pricing the European options based on expected payoff is a no-arbitrage valuation.

### 3.2. BSM Differential Equation

To derive the Black–Scholes-Merton differential equation, we start with the expected payoff equation,
(48)$$V\left(\ln S,K,t\right)=\int d\ln S_{T}\;P\left(\ln S_{T},T|\ln S,t\right)\;\left(S_{T}-K\right).\;$$
Notice that we left the bounds of the integral such that we can use it both for call and put options. Next, we take time derivative,
(49)$$\partial_{t}V=\int d\ln S_{T}\;\left(S_{T}-K\right),\;\partial_{t}P\left(\ln S_{T},T|\ln S,t\right),\;$$
where the time derivative of the transition probability is given by a Backward–Kolmogorov equation,
(50)$$\partial_{t}P\left(\ln S_{T},T|\ln S,t\right)=-\left(r_{f}-\frac{\sigma^{2}}{2}\right)\frac{\partial P(\ln S_{T},T|\ln S,t)}{\partial \ln S}-\frac{\sigma^{2}}{2}\frac{\partial^{2}P\left(\ln S_{T},T|\ln S,t\right)}{\partial {\left(\ln S\right)}^{2}}.\;$$
Substituting Equation (Equation 50) into Equation (Equation 49), we get
(51)$$\begin{array}{cc} \partial_{t}V= & \int d\ln S_{T}\;\left(S_{T}-K\right)\left[-\left(r_{f}-\frac{\sigma^{2}}{2}\right)\frac{\partial P}{\partial \ln S}-\frac{\sigma^{2}}{2}\frac{\partial^{2}P}{\partial {\left(\ln S\right)}^{2}}\right]\\ = & -\left(r_{f}-\frac{\sigma^{2}}{2}\right)\;\frac{\partial }{\partial \ln S}\;\int d\ln S_{T}\;\left(S_{T}-K\right)\;P\left(\ln S_{T},T|\ln S,t\right)\\ & -\frac{\sigma^{2}}{2}\;\frac{\partial^{2}}{\partial {\left(\ln S\right)}^{2}}\;\int d\ln S_{T}\;\left(S_{T}-K\right)\;P\left(\ln S_{T},T|\ln S,t\right)\\ = & -\left(r_{f}-\frac{\sigma^{2}}{2}\right)\;\frac{\partial V}{\partial \ln S}-\frac{\sigma^{2}}{2}\;\frac{\partial^{2}V}{\partial {\left(\ln S\right)}^{2}}. \end{array}$$
We can rewrite this equation as
(52)$$\partial_{t}V+r_{f}\;S\;\frac{\partial V}{\partial S}+\frac{\sigma^{2}\;S^{2}}{2}\;\frac{\partial^{2}V}{\partial S^{2}}=0.$$
The partial differential equation for the option premium is derived just by substituting $E=e^{-r_{f}\left(T-t\right)}\;V$ into the above equation
(53)$$\partial_{t}E+r_{f}\;S\;\frac{\partial E}{\partial S}+\frac{1}{2}\;\sigma^{2}\;S^{2},\;\frac{\partial^{2}E}{\partial S^{2}}-r_{f}\;E=0,$$
where *E* stands for both call and put options. This is the celebrated Black–Scholes–Merton equation for European options. To solve the BSM equation for put or call options, we need to apply the right boundary conditions.

## 4. Summary and Discussion

We laid down an entropic framework to model the dynamics of stocks and European options. In our formalism, the dynamical model is derived by maximizing the relative entropy subject to the information relevant to system of interest. An important contribution of our work is introducing this alternative framework, which leads to deriving a stochastic process. The Geometric Brownian Motion model of stock price is derived by taking into account two pieces of information and imposing the scaling symmetry. It is noteworthy to mention that other literature on the subject starts by assuming an ad hoc stochastic process to model the dynamics, whereas we derive the dynamics.

Next, we extended our entropic stock model to value European options on stocks. Derivative securities ought to be priced such that there is no arbitrage opportunity. To this end, we incorporated the no arbitrage, or the risk-neutral, information in our formalism and the risk-neutral probability density was derived. To value the options premium, we discounted the expected payoff the options at maturity. The resulting model is the same as the Black–Scholes model and a differential equation was derived to value options at any time, which is the Black–Scholes–Merton differential equation.

Our formalism makes the assumptions made to derive the GBM clear. Incorporating new information or relaxing the uniformity of the drift or volatility not only lead to an extension of the dynamics of stock price, but also to a new model of pricing the derivatives. Another relevant piece of information about the dynamics of a stock price is that jumps happen, which will be addressed in future work.

What if we have a different security like a Foreign Exchange? In another work, we showed that it is straightforward to extend our formalism to model the dynamics of a FX. In addition, we derive the dynamics of FX value and the corresponding Black–Scholes model for European Options, known as the Garman–Kohlhagen model, on foreign exchange [35]. It is remarkable that our framework can be easily adapted to model different securities.

An interesting extension is modeling the dynamics of a set of stocks. An important question about such system is how to construct an optimized portfolio. Markowitz’s portfolio theory of mean-variance has addressed such a question. We would like to further develop our model to address such a question. Since our model is based on taking into account information relevant to the problem, we will also be able to incorporate other information, or prior beliefs, available to private investors into account. This would lead to a modified portfolio theory [36].

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
