# Peer review of "Entropic Dynamics of Stocks and European Options"

_entropy, 2019, doi:10.3390/e21080765_

Round 1

Reviewer 1 Report

The authors developed a framework to model the dynamics of stocks and European options.

Please indicate the difference between this manuscript and Reference [28]. Although both papers applied to different targets (one for stock, and one for FX), both papers followed the same approach. Please indicate the contribution of this paper.

If possible, please add examples to improve the readability of the manuscript.

Please describe how the results can be put into practice.

There are also many grammatical errors in the manuscript, and some are listed below.

(Line 23) Comment: “…. takes ….”

(Line 26) Comment: “…. ensures ….”

(Line 50) Comment: What is “GBM”? Spell out an acronym at its first appearance.

(Line 62) Comment “Constructing …. are is …”

(Line 74) Comment: “…. provides ….”

(Line 82) Comment: “The answer originates from …

(Lines 89-90) Comment: “This is the essence iof scale invariance symmetry”

(Line 120) Comment: “….the entropy originates from …

(Line 146) “We Taylor expand the log function and keeping to …”

(Line 159) “We attain …”

Comment: Why italicize the word “attain”?

Author Response

Hello,

Thanks so much for taking the time to review the paper.

Reviewer 2 Report

This article developes an entropic framework to model the dynamics of a stock price using a simple model in which they only captured two pieces of information, namely the continuity of dynamics and the directionality.  Using the risk-neutral measure, they derive the Black-Scholes model and the BSM differential equation. Incorporating new information or relaxing the uniformity of the drift or volatility not only lead to an extension of the dynamics of stock price, but also to a new model of pricing the derivatives.  

I think that this article is potentially interesting. I hope that my comments will be of some help for authors to revise the article. 

1. There are many papers dealing with this issue therefore the novelty of this paper needs to be highlighted.

2. The literature review should be extended.

3. The results should be discussed and compared with other studies.

4. Conclusions should be improved. Policy implications should be added.

Author Response

Hello,

Thank so much for taking the time to review our paper.

Round 2

Reviewer 1 Report

The authors have responded to the reviewer's comment properly.

Other comments:

(Line 69) comment: “… by introductioning a …” (Line 247) “Our formalism makes the assumptions made to derive the GBM clear.”

        Comment: check English.

Reviewer 2 Report

I think that the article is well revised. I am glad to say that this version can be accepted for publication.